**Data Availability Statement:** The data have been included in the Supplementary file, S1 Table.

# The intersection of health and housing: Analysis of the research portfolios of the National Institutes of Health, Centers for Disease Control and Prevention, and U.S. Department of Housing and Urban Development

Liberty Walton[1]*, Elizabeth Skillen[2], Emily Mosites[3], Regina M. Bures[4], Chino Amah-Mbah[5], Maggie Sandoval[5], Kimberly Thigpen Tart[6], David Berrigan[7], Carol Star[8], Dionne Godette-Greer[9], Bramaramba Kowtha[1], Elizabeth Vogt[1], Charlene Liggins[1], Jacqueline Lloyd[1]

**1** Office of Disease Prevention, National Institutes of Health, Bethesda, Maryland, United States of America, **2** Policy Analysis and Engagement Office, Office of Policy, Performance & Evaluation, Centers for Disease Control and Prevention, Atlanta, Georgia, United States of America, **3** Office of the Deputy Director for Infectious Diseases, Centers for Disease Control and Prevention, Atlanta, Georgia, United States of America, **4** Population Dynamics Branch, *Eunice Kennedy Shriver* National Institute of Child Health and Human Development, Bethesda, Maryland, United States of America, **5** Public Health and Epidemiology Practice, Westat, Rockville, Maryland, United States of America, **6** Office of Science Coordination, Planning, and Evaluation, National Institute of Environmental Health Sciences, National Institutes of Health, Durham, North Carolina, United States of America, **7** Health Behaviors Research Branch, Division of Cancer Control and Population Sciences, National Cancer Institute, Rockville, Maryland, United States of America, **8** Office of Policy Development and Research, Program Evaluation Division, U.S. Department of Housing and Urban Development, Washington, DC, United States of America, **9** Division of Extramural Science Programs, National Institute of Nursing Research, Rockville, Maryland, United States of America

* liberty.walton@nih.gov

## Abstract

### Background

Housing is a major social determinant of health that affects health status and outcomes across the lifespan.

### Objectives

An interagency portfolio analysis assessed the level of funding invested in "health and housing research" from fiscal years (FY) 2016–2020 across the National Institutes of Health (NIH), the United States Department of Housing and Urban Development (HUD), and the Centers for Disease Control and Prevention (CDC) to characterize the existing health and housing portfolio and identify potential areas for additional research and collaboration.

### Methods/Results

We identified NIH, HUD, and CDC research projects that were relevant to both health and housing and characterized them by housing theme, health topic, population, and study

**Funding:** The authors received no specific funding for this work.

**Competing interests:** The authors have declared that no competing interests exist.

design. We organized the assessment of the individual housing themes by four overarching housing-to-health pathways. From FY 2016–2020, NIH, HUD, and CDC funded 565 health and housing projects combined. The Neighborhood pathway was most common, followed by studies of the Safety and Quality pathway. Studies of the Affordability and Stability pathways were least common. Health topics such as substance use, mental health, and cardiovascular disease were most often studied. Most studies were observational (66%); only a little over one fourth (27%) were intervention studies.

## Discussion

This review of the research grant portfolios of three major federal funders of health and housing research in the United States describes the diversity and substantial investment in research at the intersection between housing and health. Analysis of the combined portfolio points to gaps in studies on causal pathways linking housing to health outcomes. The findings highlight the need for research to better understand the causal pathways from housing to health and prevention intervention research, including rigorous evaluation of housing interventions and policies to improve health and well-being.

## Introduction

Housing is well established as a major social determinant of health, which affects health status, healthcare access, and health outcomes across the lifespan [1, 2]. Evidence of the link between housing and health is presented in the scientific literature, literature reviews, and policy briefs [3]. For instance, exposure to housing affordability stress has been shown to have a negative impact on mental health, indoor mold has been associated with incidence of asthma, people experiencing homelessness have lower rates of breast cancer screening, and low neighborhood walkability has been associated with increased risk of diabetes [4–7]. A 2018 "Housing and Health" policy brief described four pathways by which housing and health are connected [2]. The housing-to-health pathways (referred to as "housing pathways" or "pathways") consist of Stability (access to stable housing), Safety and Quality (conditions within the home), Affordability (financial burdens of high housing costs), and Neighborhood (surrounding environment and social conditions). While research has demonstrated that these pathways affect health outcomes and costs, the specific mechanisms that link housing to health are complex and not fully understood. For example, the relationship between homelessness and health is well established, but it also co-occurs with other social determinants of health including poverty, food insecurity, access to transportation, and education quality [8]. The COVID-19 pandemic highlighted housing, specifically eviction and housing displacement, as a major social determinant of health and contributor to viral spread and poorer outcomes, especially in low-income populations and communities of color [9]. Further research could contribute to better understanding the link between housing and health and inform interventions to improve health outcomes.

Agencies across the federal government recognize the importance of housing in relation to health and well-being. Healthy People 2020 and 2030, for example, include numerous, data-driven, national objectives focused on promoting healthy and safe home environments and reducing health disparities [10, 11]. The Community Preventive Services Task Force, an independent, non-federal body of public health and prevention experts, has issued several recommendations related to housing and population health [12, 13]. The important link between housing and health is recognized globally. The World Health Organization provides housing

and health guidelines that demonstrate the critical housing and health relationship globally [14]. In 2017, the Canadian federal government launched its first national, 10-year strategy to ensure Canadians have access to affordable housing [15]. In 2019, the Canadian Parliament passed the National Housing Strategy Act, which recognizes housing as a human right [16]. The United Kingdom established a long-term plan for safe and healthy homes for all in 2023, which includes a National Planning Policy and Framework that sets out the government's planning policies and how policies should be applied locally with the understanding that health depends on where people live [17, 18]. Housing continues to be a fundamental public health issue worldwide [19].

In 2019, representatives from the National Institutes of Health (NIH), the U.S. Department of Housing and Urban Development (HUD), and the Centers for Disease Control and Prevention (CDC) recognized their common interests in the intersection of housing and health and formed an interagency Health and Housing Group. Known as the nation's medical research agency, the mission of NIH is to seek fundamental knowledge to improve health and save lives [20]. This includes supporting research to understand and address social and structural factors that impact health [21]. HUD's mission is to create strong, sustainable, inclusive communities and quality affordable homes for all. Advancing sustainable communities by recognizing housing's role as essential to health is a component of HUD's FY2022-2026 strategic goals [22]. As the nation's health protection agency, CDC conducts critical science and provides health information that protects the nation against health threats and responds when they arise [23]. While there is no specific "health and housing" office at CDC, housing and homelessness are included and explored as risk or protective factors in multiple programmatic units [24–28]. Despite differences in the missions of member agencies, there are some commonalities in the research areas supported by the three agencies, and efforts to summarize and coordinate research initiatives could be beneficial to the agencies, the housing research field, and practice communities. Findings from NIH, HUD, and CDC health and housing research portfolios inform future research, U.S. policy, and public health recommendations, such as those developed by the CDC Community Guide, which produces evidence-based recommendations for public health including housing and many other national and global organizations, such as the US National Institute of Building Sciences [29, 30].

The intent of the Health and Housing Group is to foster collaborations across federal agencies to advance prevention research focused on both housing and health, particularly studies of interventions to promote equitable health and well-being. A further goal of the health and housing working group has been to discuss the wide range of outcomes and exposures salient to health and housing research and identify outcomes of common interest. Efforts to harmonize and prioritize health-related measurements are underway at NIH via the PhenX toolkit which already includes a few measures related to housing [31]. To inform the group's work and understand the landscape of research supported by the three agencies, the group conducted a portfolio analysis of research projects (also referred to as "studies") funded by NIH, HUD, and CDC between fiscal years (FY) 2016 and 2020 in which both housing and health were explicit study foci. This paper describes the health and housing research portfolios across the three agencies and reveals common themes and gaps with the intent of identifying areas for further research and opportunities for cross-federal agency collaborative efforts to advance health and housing intervention research.

## Methods

Given the paucity of information available that summarizes the health and housing research portfolio across NIH, HUD, and CDC, a descriptive portfolio analysis is necessary. This

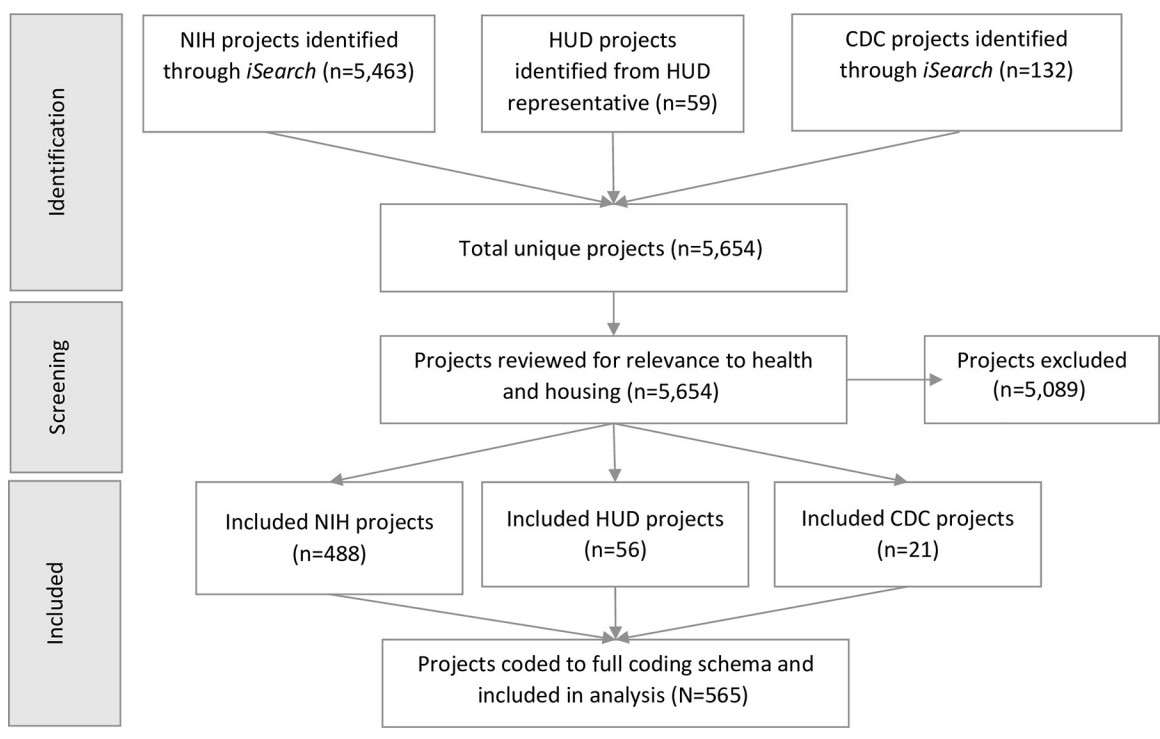

**Fig 1. PRISMA flow diagram to identify projects focused on both health and housing.**

analysis allows us to better understand the health and housing research funded across the three agencies and identify areas for future research and collaborations. We used the NIH Office of Portfolio Analysis' *iSearch Grants* platform, which pulls U.S. Department of Health and Human Services (HHS) grant data from the internal IMPACII database, to identify relevant NIH projects and CDC projects. In *iSearch*, we searched for housing-related keywords in the project titles and abstracts, including the terms 'homeless,' 'home,' 'hous*,' 'neighborhood,' and 'built environment'. To identify additional relevant NIH projects, we searched for further topics of interest using categories within NIH's Research, Condition, and Disease Categorization (RCDC) system, notably, 'Physical Activity,' 'Obesity,' 'Nutrition,' 'Asthma,' 'Social Determinants of Health,' and 'Homelessness' categories. Since CDC projects do not have RCDC codes, we identified additional CDC projects that were highly relevant to health and housing using the same key words and applied the *iSearch* 'Find Similar Projects' function. Because HUD projects are not available in *iSearch*, HUD staff provided data for HUD projects that included an explicit focus on health and housing. Our initial search yielded 5,654 unique projects for NIH (n = 5,463), HUD (n = 59), and CDC (n = 132) (Fig 1).

Next, we further refined our list of projects. First, we limited projects to new, renewal, and competing revision projects for NIH and CDC funded between FY 2016 and 2020 to obtain a picture of the funded portfolios over the five-year period prior to the convening of the work group. For multicomponent grants (e.g., center grants and cooperative agreements), we included only relevant individual research subprojects; administrative and other non-research project cores were not included. We excluded loan repayment grants and projects that were conducted solely outside of the United States. Since we used broad search criteria to identify the projects, it was necessary to further review the projects for relevance. For example, the RCDC 'Social Determinants of Health' category includes housing, but it also includes other social determinants of health. Therefore, we reviewed all projects for relevance to health and

housing. In addition, projects identified via keyword search strategies required manual review for relevance. For instance, projects that mentioned 'animal housing' were initially identified when using the keyword "hous*", but these are not relevant to our health and housing portfolio. Members of the Health and Housing group and contract analysts reviewed the available project titles, abstracts, and public health relevance statements for each project to determine which projects included a focus on both housing and health. This review resulted in a final set of 565 projects for our analysis that were relevant to both health and housing (NIH n = 488, HUD n = 56, CDC n = 21) (S1 Table).

To further characterize the projects, we developed a coding schema of categories and topics to manually apply to the projects in the portfolio analysis (S1 File). The schema included nine categories (e.g., housing theme, study design, geographic setting) with 73 non-mutually exclusive topics (e.g., built environment, homelessness, observational study, cost-benefit analysis, urban, rural) for characterizing the studies by housing theme, health topic, design, location, and population. Examples of housing themes include built environment, housing environmental exposures, neighborhood risk characteristics, segregation, homelessness, and subsidized housing (Table 1). Health topic refers to the health-related condition or subject that was studied, such as substance use, mental health, asthma, and cancer. Study design includes observational, intervention, cost-benefit, and methods development, which includes developing and testing new measures, tools, or analyses. The categories and topics were established through discussions with the Health and Housing Group, and revisions were made to better define the categories after pilot coding an initial sample of the projects. We manually applied the coding schema to the available title, abstract, and public health relevance statement for each project.

**Table 1. Housing pathways and themes.**

| Housing Pathways and Themes | Examples |
| --- | --- |
| **Neighborhood** | Surrounding environment and social conditions |
| **Built Environment** | Retail outlets, sidewalks, parks |
| **Neighborhood Socioeconomic Disadvantage** | Poverty or low wages at the neighborhood-level |
| **Neighborhood Risk Characteristics** | Neighborhood violence or crime, neighborhood social environment, neighborhood disorder |
| **Segregation** | Neighborhood or residential segregation |
| **Safety and Quality** | Conditions within the home |
| **Housing Environmental Exposures** | Air or water pollution, flooding, lead exposure, household dust, noise, and home tobacco smoke exposure |
| **Housing Remediations** | Interventions to remediate housing to improve health |
| **Affordability** | Financial burdens of high housing costs |
| **Individual Socioeconomic Disadvantage** | Individual socioeconomic disadvantage, participants of Supplemental Nutrition Assistance Program (SNAP) or Women, Infants and Children (WIC) |
| **Stability** | Consistent access to housing |
| **Homelessness** | People living in shelters |
| **Subsidized Housing** | Subsidies, public housing, housing vouchers |
| **Housing Insecurity** | Non-residential status, housing instability, home loss after disaster |
| **Other** | Transitions in living arrangements, systems of care, household firearm storage, "home environment" |

Note: Housing themes were aligned with the housing pathways to health outcomes identified in the "Housing and Health" policy brief [2]. However, for the "Safety and Quality" housing pathway, environmental exposures from both within the house, as well as from the surrounding neighborhood were included in our analysis.

We coded each grant independently and an additional coder reviewed and verified the coding. We discussed and clarified any questions related to the coding with the Health and Housing Group members. Following the coding process, we organized our housing themes by the housing pathways identified in the 2018 "Housing and Health" policy brief (Table 1) [2]. We used Microsoft 365 Excel to produce descriptive statistics.

## Results

From FY 2016–2020, the number of new health and housing projects funded each year fluctuated between 82 and 108 for NIH, six and 18 for HUD, and two and six for the CDC (Fig 2). Among the 565 NIH, HUD, and CDC projects included in the analysis, NIH funded 488 projects for approximately $231 million, HUD funded 56 projects for over $42 million, and CDC funded 21 projects for almost $10.3 million. (Note: Budget data were unavailable for one HUD in-house research project and three CDC projects). NIH's investment in health and housing research increased by 32.8% from approximately $39.1 million in 2016 to $51.9 million in 2020. For HUD, the portfolio of housing research that included a health focus increased by 155.7% from $4.4 million in 2016 to $11.3 million in 2020. CDC also had an upward trend in investments in health and housing research between 2016 and 2019 ($1.5 million to $4.6 million), with a decrease in 2020 to $1.5 million (net increase of less than 1% from 2016) (Fig 2). Although there was a decrease in new NIH health and housing projects funded in 2018, NIH funding in this area steadily increased through 2020. HUD funding in this area was at its highest in 2018.

We characterized the housing themes for each agency portfolio by the four housing pathways (Table 2). Most NIH (n = 303, 62%) and CDC projects (n = 18, 86%) focused on the Neighborhood, while HUD projects focused largely on Safety and Quality (n = 54, 96%). Across the three agencies combined, Stability was the least frequently studied pathway (n = 134, 24%).

We also summarized the housing themes within each pathway (Table 2). Within the Neighborhood pathway, NIH projects focused largely on the built environment (n = 191 of 303), while HUD (n = 6 out of 7) and CDC (n = 11 of 18) projects focused largely on neighborhood

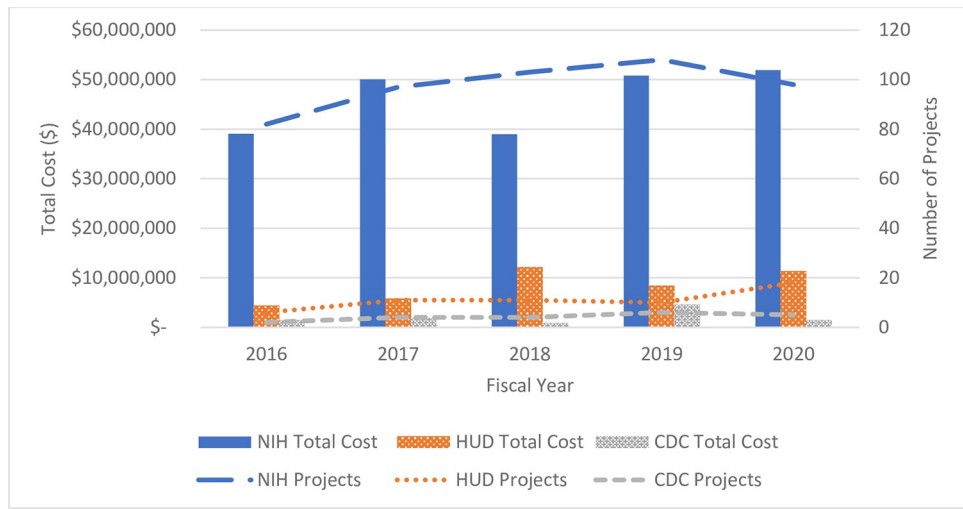

**Fig 2. Trends in health and housing funding for NIH, HUD, and CDC (FY 2016–2020).** Note: NIH, National Institutes of Health; HUD, U.S. Department of Housing and Urban Development; CDC, Centers for Disease Control and Prevention.

**Table 2. Agency portfolios organized by housing pathway and theme, health topic, population, and study design (FY 2016–2020).**

| | NIH Projects (n = 488) | HUD Projects (n = 56) | CDC Projects (n = 21) | Total Projects (N = 565) |
|---|---|---|---|---|
| **Housing Pathways and Themes** | | | | |
| **Neighborhood** | **303 (62%)[a]** | **7 (13%)** | **18 (86%)** | **328 (58%)** |
| Built environment | 191 (39%) | 1 (2%) | 7 (33%) | 199 (35%) |
| Neighborhood Socioeconomic Disadvantage | 100 (20%) | 6 (11%) | 7 (33%) | 113 (20%) |
| Neighborhood Risk Characteristics | 89 (18%) | 1 (2%) | 11 (52%) | 101 (18%) |
| Segregation | 21 (4%) | 0 (0%) | 0 (0%) | 21 (4%) |
| **Safety and Quality** | **142 (29%)** | **54 (96%)** | **3 (14%)** | **199 (35%)** |
| Housing Environmental Exposures | 133 (27%) | 43 (77%) | 3 (14%) | 179 (32%) |
| Housing Remediations | 11 (2%) | 29 (52%) | 0 (0%) | 40 (7%) |
| **Affordability** | **115 (24%)** | **25 (45%)** | **5 (24%)** | **145 (26%)** |
| Individual Socioeconomic Disadvantage | 115 (24%) | 25 (45%) | 5 (24%) | 145 (26%) |
| **Stability** | **114 (23%)** | **17 (30%)** | **3 (14%)** | **134 (24%)** |
| Homelessness | 51 (10%) | 0 (0%) | 1 (5%) | 52 (9%) |
| Subsidized Housing | 33 (7%) | 16 (29%) | 2 (10%) | 51 (9%) |
| Housing Insecurity | 43 (9%) | 1 (2%) | 2 (10%) | 46 (8%) |
| **Other** | **8 (2%)** | **0 (0%)** | **0 (0%)** | **8 (1%)** |
| **Health Topics** | | | | |
| Substance Use | 86 (18%) | 3 (5%) | 1 (5%) | 90 (16%) |
| Mental Health | 43 (9%) | 1 (2%) | 2 (10%) | 46 (8%) |
| Cardiovascular Disease/Heart Disease | 44 (9%) | 1 (2%) | 1 (5%) | 46 (8%) |
| Obesity/Sedentary | 43 (9%) | 0 (0%) | 2 (10%) | 45 (8%) |
| Physical Activity | 44 (9%) | 0 (0%) | 0 (0%) | 44 (8%) |
| HIV/AIDS | 40 (8%) | 0 (0%) | 2 (10%) | 42 (7%) |
| Asthma and/or COPD | 30 (6%) | 12 (21%) | 0 (0%) | 42 (7%) |
| Health Impacts of Housing Policy | 24 (5%) | 10 (18%) | 1 (5%) | 35 (6%) |
| Diabetes | 26 (5%) | 0 (0%) | 3 (14%) | 29 (5%) |
| Cancer | 28 (6%) | 1 (2%) | 0 (0%) | 29 (5%) |
| Violence (Domestic and Neighborhood) | 14 (3%) | 0 (0%) | 9 (43%) | 23 (4%) |
| Other Special Health Topics/Unspecified | 213 (44%) | 42 (75%) | 6 (29%) | 261 (46%) |
| **Populations** | | | | |
| Adults (ages 19+)/Age Unspecified[b] | 410 (84%) | 44 (79%) | 18 (86%) | 472 (84%) |
| Older Adults (ages 65+) | 53 (11%) | 6 (11%) | 1 (5%) | 60 (11%) |
| Children (through age 18) | 216 (44%) | 24 (43%) | 14 (67%) | 254 (45%) |
| Low-income | 173 (35%) | 27 (48%) | 6 (29%) | 206 (36%) |
| Minority Race/Ethnicity | 193 (40%) | 2 (4%) | 5 (24%) | 200 (35%) |
| Urban | 165 (34%) | 17 (30%) | 11 (52%) | 193 (34%) |
| People Experiencing Homelessness | 51 (10%) | 0 (0%) | 1 (5%) | 52 (9%) |
| Rural | 42 (9%) | 4 (7%) | 0 (0%) | 46 (8%) |
| Pregnant Persons | 38 (8%) | 0 (0%) | 1 (5%) | 39 (7%) |
| Sexual and Gender Minorities | 14 (3%) | 0 (0%) | 1 (5%) | 15 (3%) |
| Incarcerated | 16 (3%) | 0 (0%) | 0 (0%) | 16 (3%) |
| Persons with Disabilities | 10 (2%) | 4 (7%) | 0 (0%) | 14 (2%) |
| Immigrants/Refugees | 13 (3%) | 0 (0%) | 0 (0%) | 13 (2%) |
| Military/Veterans | 5 (1%) | 1 (2%) | 1 (5%) | 7 (1%) |
| Foster Care | 2 (0%) | 1 (2%) | 0 (0%) | 3 (1%) |
| **Study Designs** | | | | |
| Observational | 344 (70%) | 18 (32%) | 11 (52%) | 373 (66%) |

*(Continued)*

**Table 2.** (Continued)

|  | NIH Projects (n = 488) | HUD Projects (n = 56) | CDC Projects (n = 21) | Total Projects (N = 565) |
|---|---|---|---|---|
| Intervention | 120 (25%) | 27 (48%) | 8 (38%) | 155 (27%) |
| Methods Development | 56 (11%) | 10 (18%) | 4 (19%) | 70 (12%) |
| Cost-benefit | 14 (3%) | 16 (29%) | 1 (5%) | 31 (5%) |
| Other/Unclear Design | 39 (8%) | 13 (23%) | 4 (19%) | 56 (10%) |
| **Total Projects** | **488** | **56** | **21** | **565** |

Note: Housing themes, health topics, populations, and study designs are not mutually exclusive (i.e., one project could include multiple housing themes and/or health topics and/or populations and/or study designs). NIH, National Institutes of Health; HUD, U.S. Department of Housing and Urban Development; CDC, Centers for Disease Control and Prevention.

[a] Percentages indicate column percentages of total projects by each agency.

[b] This category includes projects that focused on adults, plus any projects for which an age was not specified in the abstract and there was no indication of a sole focus on children in the study.

socioeconomic disadvantage and neighborhood risk characteristics, respectively. Segregation was the least frequently studied housing theme within the Neighborhood pathway across the three agency portfolios (NIH n = 21 of 303, HUD n = 0 of 7, and CDC n = 0 of 18). Looking at the Safety and Quality pathway, housing environmental exposures (n = 179, 32%) were studied more often than housing remediations (n = 40, 7%) across the agency portfolios. The Affordability pathway solely includes the individual socioeconomic disadvantage theme, which pertained to 26% (n = 145) of projects combined across the three agencies. Finally, within the Stability pathway, NIH projects largely focused on homelessness (n = 51 of 114) and housing insecurity (n = 43 of 114). HUD projects largely focused on subsidized housing (n = 16 of 17). For the most part, CDC projects were evenly distributed across all three of the Stability housing themes (n ranged from 1 to 2 of 3). Overall, segregation, housing remediation, housing insecurity, homelessness, and subsidized housing were among the least frequently studied housing themes within the combined agency portfolios which had health as an explicit study focus.

We looked at specific health topics studied within the projects (Table 2). Almost one fifth of NIH projects examined substance use (i.e., drug, alcohol, and tobacco use) (n = 86, 18%), followed by a relatively even distribution across mental health, cardiovascular disease, obesity, physical activity, and HIV/AIDS. Almost a quarter of HUD projects focused on asthma and/or chronic obstructive pulmonary disease (COPD) (n = 12, 21%), followed by the health impacts of housing policy (n = 10, 18%). Violence, including domestic and neighborhood violence, was the most common health topic for CDC projects (n = 9, 43%). However, across all three agencies combined, violence was among the least frequently studied health topics, along with cancer and diabetes.

Table 2 characterizes the study populations within each of the agency's portfolios. Across the three agencies, an overwhelming majority of projects focused on adults, or the population was not clearly defined (n = 472, 84%). Only 11% of the total projects, however, focused specifically on adults ages 65 and older (n = 60). Fewer than half of the projects focused specifically on children (n = 254, 45%). More than a third of the projects included low-income populations (n = 206, 36%) or minority racial/ethnic groups (n = 200, 35%). Each of the remaining populations were the focus in less than 10% of the portfolio. This includes individuals experiencing homelessness, pregnant persons, sexual and gender minorities, incarcerated populations, persons with disabilities, immigrants/refugees, military/veteran populations, and children in foster care. With respect to research setting, projects more often occurred in urban settings (n = 193, 34%), with a much smaller number in rural settings (n = 46, 8%).

**Housing Pathways and Themes**

| Health Topics | Neighborhood (n=328) | | | | Safety and Quality (n=199) | | Affordability (n=145) | Stability (n=134) | | | Other Housing Themes (n=8) |
|---|---|---|---|---|---|---|---|---|---|---|---|
| | Built Environment (n=199) | Neighborhood Socioeconomic Disadvantage (n=113) | Neighborhood Risk Characteristics (n=101) | Segregation (n=21) | Housing Environmental Exposures (n=179) | Housing Remediations (n=40) | Individual Socioeconomic Disadvantage (n=145) | Homelessness (n=52) | Subsidized Housing (n=51) | Housing Insecurity (n=46) | |
| Substance Use (n=90) | 27 | 18 | 19 | 2 | 12 | 5 | 24 | 25 | 13 | 7 | 2 |
| Mental Health (n=46) | 15 | 10 | 8 | | 8 | 1 | 9 | 9 | 4 | 7 | 2 |
| Cardiovascular Disease (n=46) | 24 | 12 | 12 | 2 | 16 | | 8 | 1 | 1 | 3 | |
| Obesity (n=45) | 31 | 11 | 9 | 3 | 10 | 2 | 13 | | 2 | 1 | |
| Physical Activity (n=44) | 35 | 8 | 6 | | 8 | 3 | 10 | 1 | 4 | | |
| HIV/AIDS (n=42) | 12 | 6 | 9 | | 1 | 1 | 7 | 15 | 4 | 9 | |
| Asthma/COPD (n=42) | 11 | 5 | | | 39 | 8 | 15 | | 5 | | |
| Diabetes (n=29) | 18 | 4 | 8 | | 7 | | 7 | 1 | 1 | 1 | |
| Cancer (n=29) | 11 | 4 | 13 | 7 | 9 | | 3 | | 1 | 2 | |
| Violence (n=23) | 6 | 9 | 14 | 3 | | | 3 | 1 | 1 | 1 | 1 |

**Fig 3. Projects by health topic, housing pathway, and theme (FY 2016–2020).** Note: Health topics and housing themes are not mutually exclusive (i.e., one project could have multiple health topics and/or housing themes). 193 projects did not study one of the listed health topics and are not included in this figure.

As shown in Table 2, study designs varied across the three agencies. Across all three agency portfolios combined, observational studies were most common (n = 373, 66%), followed by intervention studies, including randomized controlled trials and non-randomized study designs such as natural and quasi-experiments (n = 155, 27%). The overwhelming majority of NIH projects were observational (70%), followed by intervention studies (25%). In contrast, almost half of HUD projects were intervention studies (48%), followed by observational studies (32%). A little over half of CDC projects were observational studies (52%), followed by more than a third intervention studies (38%). The least frequent study types were methods development, followed by cost-benefit studies. Almost one third of HUD projects were cost-benefit studies. It is important to note that our coding schema specifically coded for cost-benefit studies and may have missed studies that include other health economic analyses and methods, which are integral to agencies' overall missions to improve health [32–34].

To highlight the health and housing research topics funded across the three agency portfolios (N = 565), Fig 3 presents a cross comparison of housing pathways, themes, and health topics. The most common intersection was among projects that focused on *housing environmental exposure* (Safety and Quality pathway) and *asthma or COPD* (n = 39, 7%). The next most common were projects on the *built environment* (Neighborhood pathway) and the health topics *physical activity* (n = 35, 6%), *obesity* (n = 31, 5%), and *substance use* (n = 27, 5%). Studies on the intersection of some health topics and housing themes—for example the effect of segregation on mental health outcomes—were not prevalent in this analysis.

We also conducted a cross comparison of the combined portfolio by populations and housing pathways (Table 3). For projects that focused on adults/age unspecified and for those that focused on children, the two most common housing pathways were Neighborhood and Safety and Quality. Neighborhood was the most often studied housing pathway across most populations. Studies of low-income populations most often focused on Affordability; studies of people experiencing homelessness, incarcerated individuals, or individuals in foster care most

**Table 3. Projects by population and housing pathway (FY 2016–2020).**

| | Neighborhood (n = 328, 58%)[a] | Safety and Quality (n = 199, 35%) | Affordability (n = 145, 26%) | Stability (n = 134, 24% | Other (n = 8, 1%) | Total (N = 565) |
|---|---|---|---|---|---|---|
| **Adults (ages 19+)/Age Unspecified (n = 472, 84%)** | 276 (58%)[b] | 161 (34%) | 121 (26%) | 120 (25%) | 6 (1%) | 472 |
| **Children (through age 18) (n = 254, 45%)** | 145 (57%) | 98 (39%) | 66 (26%) | 54 (21%) | 4 (2%) | 254 |
| **Low-income (n = 206, 36%)** | 86 (42%) | 62 (30%) | 144 (70%) | 108 (52%) | (0%) | 206 |
| **Minority Race/Ethnicity (n = 200, 35%)** | 160 (80%) | 46 (23%) | 55 (28%) | 24 (12%) | 2 (1%) | 200 |
| **Urban (n = 193, 34%)** | 128 (66%) | 66 (34%) | 53 (27%) | 42 (22%) | (0%) | 193 |
| **People Experiencing Homelessness (n = 52, 9%)** | 8 (15%) | 4 (8%) | 10 (19%) | 52 (100%) | (0%) | 52 |
| **Rural (n = 46, 8%)** | 24 (52%) | 19 (41%) | 12 (26%) | 5 (11%) | 1 (2%) | 46 |
| **Pregnant Persons (n = 39, 7%)** | 24 (62%) | 22 (56%) | 10 (26%) | 2 (5%) | (0%) | 39 |
| **Sexual and Gender Minorities (n = 15, 3%)** | 12 (80%) | (0%) | 1 (7%) | 3 (20%) | (0%) | 15 |
| **Incarcerated (n = 16, 3%)** | 5 (31%) | 3 (19%) | 5 (31%) | 13 (81%) | (0%) | 16 |
| **Persons with disabilities (n = 14, 2%)** | 6 (43%) | 8 (57%) | 3 (21%) | 3 (21%) | (0%) | 14 |
| **Immigrants/Refugees (n = 13, 2%)** | 12 (92%) | (0%) | 1 (8%) | 2 (15%) | (0%) | 13 |
| **Military/Veterans (n = 7, 1%)** | 4 (57%) | 1 (14%) | (0%) | 2 (29%) | (0%) | 7 |
| **Foster Care (n = 3, 1%)** | 1 (33%) | 1 (33%) | 1 (33%) | 3 (100%) | (0%) | 3 |

Note: Populations and housing themes are not mutually exclusive (i.e., one project could have multiple populations and/or housing themes).

[a]Percentages in the housing pathway and population headers indicate the percentage of total projects.

[b]Table percentages indicate row percentages.

often focused on Stability; and projects studying persons with disabilities most often focused on Safety and Quality. Some populations and housing pathways did not intersect such as sexual and gender minorities and housing Safety and Quality, and military/veterans and Affordability. More details on the populations and housing pathways and themes are in the S2 Table.

Finally, Table 4 summarizes project study designs organized by the housing pathways and themes. Overall, observational studies (n = 373, 66%) were the most common study design across most pathways. Intervention studies were most frequent for studies that focused on housing remediations (Safety and Quality pathway), homelessness (Stability pathway), and subsidized housing (Stability pathway). Intervention study designs were less common in projects focused on the Neighborhood pathway (i.e., built environment, neighborhood socioeconomic disadvantage, neighborhood risk characteristics, or segregation). Methods development (n = 70, 12%) and cost-benefit studies (n = 31, 5%) were the least frequent across most housing themes.

## Discussion

Funding for research at the intersection of housing and health is spread across various federal agencies, including NIH, HUD, and CDC. This analysis characterizes new research projects funded by these agencies during FY 2016–2020 that include an explicit focus on housing and health topics. The analysis revealed the following features of the health and housing portfolios across the three agencies: 1) funding investment increased over time; 2) a majority of projects used observational study designs; 3) projects focused on a broad array of themes across the four housing pathways, ranging from the built environment and environmental exposures to

**Table 4. Projects by study design, housing pathways, and themes (FY 2016–2020).**

| Housing Pathways and Themes | Observational (n = 373, 66%)[a] | Intervention (n = 155, 27%) | Methods Development (n = 70, 12%) | Cost-benefit (n = 31, 5%) | Other/Unclear Design (n = 56, 10%) | Total (N = 565) |
|---|---|---|---|---|---|---|
| **Stability** | | | | | | |
| **Housing Insecurity (n = 46, 8%)** | 27 (59%)[b] | 17 (37%) | 8 (17%) | 1 (2%) | 8 (17%) | 46 |
| **Subsidized Housing (n = 51, 9%)** | 20 (39%) | 29 (57%) | 2 (4%) | 6 (12%) | 6 (12%) | 51 |
| **Homelessness (n = 52, 9%)** | 17 (33%) | 30 (58%) | 6 (12%) | 2 (4%) | 12 (23%) | 52 |
| **Safety and Quality** | | | | | | |
| **Housing Remediations (n = 40, 7%)** | 5 (13%) | 25 (63%) | 3 (8%) | 9 (23%) | 11 (28%) | 40 |
| **Housing Environmental Exposures (n = 179, 32%)** | 121 (68%) | 47 (26%) | 27 (15%) | 16 (9%) | 13 (7%) | 179 |
| **Affordability** | | | | | | |
| **Individual Socioeconomic Disadvantage (n = 145, 26%)** | 80 (55%) | 57 (39%) | 10 (7%) | 14 (10%) | 14 (10%) | 145 |
| **Neighborhood** | | | | | | |
| **Built environment (n = 199, 35%)** | 145 (73%) | 44 (22%) | 30 (15%) | 5 (3%) | 10 (5%) | 199 |
| **Segregation (n = 21, 4%)** | 18 (86%) | 3 (14%) | 3 (14%) | 0 (0%) | 1 (5%) | 21 |
| **Neighborhood Risk Characteristics (n = 101, 18%)** | 86 (85%) | 10 (10%) | 16 (16%) | 1 (1%) | 5 (5%) | 101 |
| **Neighborhood Socioeconomic Disadvantage (n = 113, 20%)** | 82 (73%) | 27 (24%) | 10 (9%) | 3 (3%) | 7 (6%) | 113 |
| **Other (n = 8, 1%)** | 8 (100%) | 1 (13%) | 0 (0%) | 0 (0%) | 1 (13%) | 8 |

Note: Housing themes and study designs are not mutually exclusive (i.e., one project could have multiple housing themes and/or multiple study designs).

[a]Percentages in the study design and housing theme headers indicate the percentage of total projects.

[b]Table percentages indicate row percentages.

neighborhood socioeconomic disadvantage; 4) there was lower representation of projects focused on children and on older adults over age 65, as well as lower representation for studies focused on people experiencing homelessness, sexual and gender minorities, and persons with disabilities; 5) the most studied health outcomes were substance use, mental health, and cardiovascular disease; and 6) few projects focused on housing policy evaluation and policy implementation research, especially in relation to segregation, housing insecurity, and homelessness. The findings from this analysis highlight potential gap areas for researchers to explore and opportunities for and potential benefits of collaboration and coordination among federal agencies with common interests and investments in health and housing research.

## Funding investment and study design

While not a new area of study, research on the intersection of health and housing is an evolving area that has received increased attention in recent years. The portfolio analysis shows a stable-to-gradually increasing investment in health and housing research over time across NIH, HUD, and CDC, with a few fluctuations. Comparing FY 2016 to FY 2020, the three agencies increased their yearly combined number of projects by 34% (90 to 121) with a corresponding 44% increase in funding ($45 million to $64.7 million).

Most studies in the portfolio across the three agencies were observational and non-experimental. However, the majority of studies on subsidized housing, homelessness, and housing remediations were intervention studies. Notably, only a third of studies on homelessness were observational. Recent research suggests that additional longitudinal, epidemiologic studies are needed to understand the causal mechanisms of poor health among persons experiencing

homelessness [35, 36]. In addition, intervention studies using rigorous experimental designs are needed to build the evidence base for interventions to address the multiple pathways between housing and health. Examples of such designs include cluster randomized trials and non-randomized controlled experimental designs, which have been used to evaluate interventions at the community level and in field experiments. Use of such designs could also help advance research to eliminate health disparities and advance health equity.

As indicated by the funding patterns, some housing themes lend themselves to intervention studies, such as housing remediation, homelessness, and subsidized housing. By contrast, topics such as segregation and built environment can be more difficult to study with randomized intervention trials. A potential alternative is 'natural experiment' study designs of a specific policy or environmental change, which could strengthen causal inference [37]. NIH has called for research using such approaches and funded several housing and health natural experiment studies [38, 39].

Equally important is research on implementation strategies to increase adoption, scalability, and impact of evidence-based strategies and interventions. Also important are cost-benefit studies to inform policy and program implementation decisions [40, 41]. Advancing research to build the evidence base to address the multifaceted and complex relationship between housing and health requires concerted, intentional, and collaborative efforts across diverse sectors and disciplines and presents opportunities for coordinated cross-agency efforts to advance research, programming, and policy.

## Housing pathways

Our analysis showed that, NIH and CDC projects predominantly focused on the Neighborhood pathway, with a major focus on the built environment. There is clear evidence that built environment characteristics such as sidewalks, grocery stores, green spaces, parks, and playgrounds can be protective and improve overall well-being and health outcomes, including for conditions such as cardiovascular disease, diabetes, and obesity [2, 42, 43]. Within this pathway, over half of CDC's portfolio comprised research to evaluate neighborhood risk characteristics such as violence (Table 2).

The Safety and Quality of housing was studied in just over a third of the projects in the combined portfolio of the agencies. Notably, older housing with deteriorating paint can be found in many low-income areas and may serve as a source of lead exposure [44]. In areas with concentrated disadvantage (e.g., residential segregation, concentrated poverty, and neighborhood disinvestment), housing and neighborhood characteristics may interact to increase residential exposure to environmental toxins, including lead [45]. Other home safety hazards may lead to falls and associated injury [46]; fall prevention was studied infrequently across the portfolios. Given the known links between falls and health outcomes, evidence-based interventions to prevent falls within the home environment is an area for additional research [47] that could benefit from coordinated cross-agency efforts.

A substantial portion of the health and housing portfolios for all three agencies addressed Affordability. Leveraging this alignment could provide impetus for coordinated efforts, including cross-agency funding efforts and multi-disciplinary collaboration among health and housing researchers.

The housing Stability pathway had the fewest number of health and housing projects across the three agencies combined. Specifically, our analysis identified homelessness and housing insecurity as a gap area in the overall health and housing portfolio. A recent report by the National Academies of Sciences, Engineering, and Medicine recognized similar gaps and identified the need for quantitative research to accurately depict U.S. housing insecurity, qualitative

research to identify short-term predictors of housing insecurity, and effective strategies and ordinances to mitigate eviction risk for populations that experience housing instability [48]. In addition to social factors, research is also needed to determine how issues such as climate change and increases in weather-related disasters contribute to housing insecurity and related effects on health.

## Populations

While the portfolio analysis shows projects in various populations, representation was unequal for certain groups. Most studies focused on adults or did not specify an age, while fewer than half of the projects specified the inclusion of children and approximately one tenth focused on adults over age 65. While slightly more than one third of the portfolio focused on low-income, minority racial/ethnic groups, or urban populations, less than one tenth of the portfolio focused on homeless, rural, or pregnant populations. Even fewer projects included other groups such as sexual and gender minorities, incarcerated individuals, persons with disabilities, immigrants, and children in foster care. Even for studies within the Neighborhood and Safety and Quality pathways, which had the highest representation in the portfolio, certain populations were underrepresented. For example, less than a tenth of Safety and Quality projects focused on rural populations. A recent systematic review found similar gaps in the literature for research on the intersection of housing and health in specific populations including older adults, children, and women, and identified the need for more inclusive research that explores the specific health needs of diverse populations and communities [49].

## Health topics

The five most common health-related topics studied in the portfolio were substance use, mental health, cardiovascular disease, obesity, and physical activity. The increased attention to substance use and mental health brought about by the opioid epidemic, as well as increased acknowledgment of lack of mental health and economic parity, likely contributed to increased investments in research in these areas [50–52]. Looking at intersections between housing pathways and health topics (Fig 3), the most common topics occurred between housing environmental exposures and asthma and/or COPD within Safety and Quality pathway and between the built environment and physical activity, obesity, substance use, and cardiovascular disease in the Neighborhood pathway. The analysis revealed sparce focus on chronic stress, even though housing insecurity is an adverse childhood experience (ACE) linked to stress and has been found to play a role in health outcomes [53–56].

## Policy implications

Housing policy evaluation and policy implementation research are of growing interest within the public health research community and are particularly important in this context because of the complex policy landscape pertaining to housing in the United States [57, 58]. Across the portfolios of the three agencies, there are differential emphases. For example, NIH supports policy evaluation research, but does not create housing policies. While each agency can only fund research within their specific missions and mandates, we can collaborate across agencies to complement each other's efforts.

Our analysis also revealed overall limited funding for research on segregation, an important structural determinant of health. Emerging evidence suggests that housing related policies such as redlining, a discriminatory process used historically by mortgage lenders to control where minorities could live, can have enduring effects on health disparities [43, 59, 60]. A number of studies are also identifying adverse health consequences of gentrification and

resulting physical and cultural displacement [61–63]. Recent studies document adverse impacts on older Black adults in Portland, Oregon including discrimination against Black home ownership, displacement, race-related stress, and financial burden of aging in place [64]. Studies in the literature have identified a critical need for more research exploring how structural racism continues to shape housing inequities and health disparities [59].

Worldwide, attention is being paid to the social determinants of health. A body of literature has emerged supporting the integration of policy with housing and health research, for example pertaining to urban and territorial planning, zoning, and land use policy (e.g. UN-Habitat and World Health Organization 2020) [65]. This literature emphasizes how urban planning for health as well as design issues addressed by building codes, engineering, and advances in architecture are needed to address housing and health issues such as exposure to light at night, noise, toxic indoor air exposures, and lack of access to green space, transit, and other necessities for health [14]. The WHO 2018 report notes strong evidence for efforts, including policies, to prevent crowding, support adequate temperature regulation, include safety devices such as smoke alarms, and support accessibility for people with mobility limitations. More research is needed to understand specific needs and best implementation practices in diverse settings, but affordability is an issue worldwide [66].

In 2018, the Government Accountability Office (GAO) examined collaboration between HUD and HHS, of which NIH and CDC are operating divisions, pertaining to housing for older adults and health and noted that future collaborations would benefit from defining common outcomes, which would help inform Congress on areas for focus [67]. Motivated by the clear connection between health and housing, HUD aims to expand partnerships and collaborations with HHS agencies including NIH and CDC [67]. In addition to funding research through grants and contracts, HUD has expanded access to HUD administrative data to support health and housing research through Data License Agreements with research organizations and data linkages with the National Center for Health Statistics (NCHS), which links national health surveys to HUD's administrative data [68, 69].

Addressing housing as a key social determinant of health is a priority across HHS. This is reflected in Healthy People 2030's emphasis on "reducing health and safety risks in homes" [70]. Healthy People includes data driven national housing-related objectives, including to increase the proportion of smoke-free homes, reduce blood-lead levels of children ages 1–5, address housing affordability, and increase the proportion of homeless adults with mental health problems who receive mental health services [70].

Both individual-level and structural strategies, such as policies, can help people stay safe and healthy at home. The research of these three agencies can provide crucial evidence for program and policy decisions. Collaborative efforts across federal agencies and disciplines are imperative to reduce housing-related health and safety risks and improve health for all.

## Limitations

This analysis is subject to certain limitations. The review was limited to examination of project titles, abstracts, and public health relevance statements, and relied only on information available in these sources. Future studies could review full applications to extract additional information such as geographical coverage and further details concerning demographic characteristics of study populations. We did not review the outcomes of funded projects. Areas with substantial funding may or may not produce results to guide further research and policy action. Projects may have been mischaracterized due to the nature of manual coding or any limitation of the identified search terms. To reduce mischaracterization, we used multiple search strategies and clearly defined terms of interest. However, future analyses might include

a broader set of search terms, such as "apartments" or "shelters" for the term "housing," to potentially capture a wider set of projects. Future coding schemas could also include additional topics for more detailed categorization of the research projects, for example including topics such as "health economic analyses". Our analysis focused only on projects that were new awards between FY 2016 and 2020. New projects funded prior to FY 2016 may include ongoing activities related to health and housing that are not captured in our analysis. In addition, since this portfolio was predominantly NIH-funded and NIH is made up of 27 Institutes, Centers, and Offices (ICOs), it may be more informative to conduct future analyses at the NIH ICO-level to better understand differences in funding across topics and study designs.

This analysis focused on research funded by the three U.S. federal agencies represented on the Health and Housing Group and on research conducted in the United States. Future analyses could include additional U.S. federal funders, such as the Department of Energy and the U.S. Environmental Protection Agency, among others, to provide a more comprehensive picture of health and housing research portfolios across federal agencies. For example, EPA has made significant investments in research on smart growth and affordable housing, which is not reflected in this analysis [71]. It also would be beneficial for future analyses to examine funding of health and housing research globally.

Lastly, this paper presents a descriptive analysis of the NIH, HUD, and CDC funding for housing and health research. Future studies could consider additional analyses involving more complex models to describe underlying mechanisms linking housing and health to further identify research needs in this area and could also consider more complex approaches to describing extant funding, such as machine learning, concept mapping, weighted analysis, and other statistical tools [72–74].

## Conclusions

There is increased recognition of the relationship between housing and health, particularly with respect to health outcomes and consequences associated with a lack of access to stable, high-quality, safe, and affordable housing. The disproportionate impacts of the COVID-19 pandemic and climate change disasters on communities of color have increased recognition of how social determinants of health, including housing and neighborhood characteristics, can increase the prevalence and severity of illnesses, diseases, and injuries [75, 76]. Despite a robust body of literature documenting the relationship between housing and health outcomes, questions about the mechanisms that link housing and health remain unanswered. Research to explore these questions and intervention studies utilizing a range of methodologies, including experimental and quasi-experimental study designs, could inform strategies to address disparities related to this critical social determinant and thereby improve health [2, 48]. This analysis identifies key areas for enhanced coordination across NIH, HUD, and CDC to advance research on health and housing. Coordinated and collaborative efforts across all relevant federal agencies could provide a more holistic, efficient, and impactful approach to better understand and address the complex relationships between health and housing [49].

## Supporting information

**S1 File. Coding schema.**
(DOCX)

**S1 Table. NIH, HUD, and CDC health and housing portfolios (FY 2016–2020).**
(XLSX)

**S2 Table. Projects by population, housing pathways and themes (FY 2016–2020).**
(XLSX)

## Acknowledgments

We thank the Westat and National Institutes of Health (NIH) portfolio coding team and all the members of the Health and Housing Group, which includes representatives from the NIH, the U.S. Department of Housing and Urban Development (HUD), and the Centers for Disease Control and Prevention (CDC). This article was written by employees of the NIH, HUD, CDC, and Westat; however, the statements, opinions, and conclusions in this report are those of the authors and do not necessarily represent the official position of the NIH, HUD, CDC, or any of their affiliated institutions or agencies.

## Author Contributions

**Conceptualization:** Liberty Walton, Elizabeth Skillen, Emily Mosites, Regina M. Bures, Kimberly Thigpen Tart, David Berrigan, Carol Star, Dionne Godette-Greer, Bramaramba Kowtha, Elizabeth Vogt, Charlene Liggins, Jacqueline Lloyd.

**Data curation:** Liberty Walton, Chino Amah-Mbah, Maggie Sandoval, David Berrigan, Charlene Liggins.

**Formal analysis:** Liberty Walton, Chino Amah-Mbah, Maggie Sandoval, Charlene Liggins.

**Investigation:** Liberty Walton, Elizabeth Skillen, Chino Amah-Mbah, Maggie Sandoval, Charlene Liggins, Jacqueline Lloyd.

**Methodology:** Liberty Walton, Elizabeth Skillen, Emily Mosites, Regina M. Bures, Chino Amah-Mbah, Maggie Sandoval, Kimberly Thigpen Tart, David Berrigan, Carol Star, Dionne Godette-Greer, Bramaramba Kowtha, Elizabeth Vogt, Charlene Liggins, Jacqueline Lloyd.

**Project administration:** Liberty Walton.

**Supervision:** Charlene Liggins, Jacqueline Lloyd.

**Validation:** Liberty Walton, Chino Amah-Mbah, Maggie Sandoval, Charlene Liggins.

**Visualization:** Liberty Walton, Emily Mosites, Chino Amah-Mbah, Maggie Sandoval, Charlene Liggins.

**Writing – original draft:** Liberty Walton, Elizabeth Skillen, Emily Mosites, Regina M. Bures, Maggie Sandoval, Carol Star, Charlene Liggins, Jacqueline Lloyd.

**Writing – review & editing:** Liberty Walton, Elizabeth Skillen, Emily Mosites, Regina M. Bures, Chino Amah-Mbah, Kimberly Thigpen Tart, David Berrigan, Carol Star, Dionne Godette-Greer, Bramaramba Kowtha, Elizabeth Vogt, Charlene Liggins, Jacqueline Lloyd.

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
