## [Decision Letter · Decision Letter 0]

10 Jul 2023

PONE-D-23-11897The intersection of health and housing: Analysis of the research portfolios of the National Institutes of Health, Centers for Disease Control and Prevention, and U.S. Department of Housing and Urban DevelopmentPLOS ONE

Dear Dr. Walton,

Thank you for submitting your manuscript to PLOS ONE. After careful consideration, we feel that it has merit but does not fully meet PLOS ONE’s publication criteria as it currently stands. Therefore, we invite you to submit a revised version of the manuscript that addresses the points raised during the review process.

 Please submit your revised manuscript by Aug 24 2023 11:59PM. If you will need more time than this to complete your revisions, please reply to this message or contact the journal office at plosone@plos.org. Please include the following items when submitting your revised manuscript:A rebuttal letter that responds to each point raised by the academic editor and reviewer(s). You should upload this letter as a separate file labeled 'Response to Reviewers'.A marked-up copy of your manuscript that highlights changes made to the original version. You should upload this as a separate file labeled 'Revised Manuscript with Track Changes'.An unmarked version of your revised paper without tracked changes. You should upload this as a separate file labeled 'Manuscript'.If applicable, we recommend that you deposit your laboratory protocols in protocols.io to enhance the reproducibility of your results. Protocols.io assigns your protocol its own identifier (DOI) so that it can be cited independently in the future. For instructions see: https://journals.plos.org/plosone/s/submission-guidelines#loc-laboratory-protocols. Additionally, PLOS ONE offers an option for publishing peer-reviewed Lab Protocol articles, which describe protocols hosted on protocols.io. Read more information on sharing protocols at https://plos.org/protocols?utm_medium=editorial-email&utm_source=authorletters&utm_campaign=protocols.

We look forward to receiving your revised manuscript.

Kind regards,

Alicia Chang, M.D., M.S.

Academic Editor

PLOS ONE

Journal Requirements:

**Additional Editor Comments:**

The work presented is of high interest to the journal, and we encourage you to review the comments and suggestions to further enhance the manuscript. Please respond to the reviewers' comments, and in particular, please address the questions around the policy context of the research portfolios evaluated.

Reviewers' comments:

Reviewer's Responses to Questions

**Comments to the Author**

1. Is the manuscript technically sound, and do the data support the conclusions?

Reviewer #1: Yes

Reviewer #2: Partly

2. Has the statistical analysis been performed appropriately and rigorously? 

Reviewer #1: N/A

Reviewer #2: Yes

3. Have the authors made all data underlying the findings in their manuscript fully available?

Reviewer #1: No

Reviewer #2: Yes

4. Is the manuscript presented in an intelligible fashion and written in standard English?

Reviewer #1: Yes

Reviewer #2: Yes

5. Review Comments to the Author

Reviewer #1: The authors present a systematic review of research grants awarded to investigate the intersection of housing and health by three federal agencies over a period of five years. Their explicit research question is what types of research at the intersection of health and housing are currently being funded and what opportunities exist for additional research and collaboration (ie, what areas are underfunded, how can the agencies use grantmaking as a tool of collaboration). The implicit underlying question is whether public research dollars are being spent in the best possible way to improve the health of the U.S. population.

The approach employed in the review is to identify existing research by searching databases of studies funded by the NIH, the CDC, or HUD; classify each study based on themes, broad categories (“pathways”), populations of interest, types of study, etc; and quantify the studies based on those classifications to identify patterns and gaps in the funded portfolios.

The paper is somewhat unusual, in that the authors are employees of the funding organizations whose portfolios they evaluate. So, this is essentially an internal study that is seeking external publication, either in the name of transparency, in an attempt to influence the leadership or culture of the authors' home institutions, or simply to persuade the grant-seeking community to propose new projects that fill the gaps identified. Or all of the above. Based on this context, the introduction and discussion would be strengthened by a discussion of the role of public funding in general and of the three member agencies of the Health and Housing Group in particular in driving actionable research. What proportion of current health and housing policy research is funded by these agencies--and what effects do their grantmaking priorities have upon the research topics chosen across the policy research landscape writ large?

I would say this is the overall substance of my comments below: The authors observe patterns and gaps in a data set that are not random, but rather are the result of institutional policies and priorities enforced by human choices. The authors are, of course, in a slightly difficult position in that PLOS ONE accepts only systematic reviews that are based on reproducible quantitative analysis, so too much policy discussion would be inappropriate. Too little, however, fails to capture the causal mechanisms that influenced the patterns summarized in this paper, making it harder to fully grasp the most realistic levers available to close the portfolio gaps described. I recommend accepting the manuscript once some additional policy context is added--and once the data are made available. (The submitted manuscript specifies that the data "will only be available after acceptance." I'm not sure how this comports with the PLOS Data Policy.)

Conclusions: The authors determine that the existing portfolios could be strengthened if more studies were funded that a) sought to identify the specific causal mechanisms that link housing to health, b) evaluated the effectiveness of specific interventions, or c) otherwise provided concrete guidance as to how programs and policies could improve health by increasing the quality of, or access to, housing.

Methods: Methods are well described. Categorization studies of this kind can never reach a level of 100% objectivity, so they can never be 100% reproducible, but the methods discussion convincingly indicates that subjectivity was minimized to the extent possible, and it provides sufficient information for other researchers to attempt to reproduce the results or to apply analogous methods to other agencies’ portfolios.

Results: Following from my comment above about policy context, the analysis included in the manuscript's Results section would be strengthened if the Introduction included a brief overview of each agency’s policy role and agenda, which naturally informs its research interests and determines both the opportunities and the limits of the agencies’ ability to collaborate. The inspiration for this comment is Table 2, which demonstrates fairly clearly, for example, that HUD tends to fund research only into the specific types of housing resources that it creates or regulates, rather than the broader context in which those resources exist or intervene. (eg, 96% of HUD-funded research concerns the stability and quality of housing, while 0% concerns the health effects of homelessness and 2% concerns the built environment.) Does that narrow focus represent an opportunity, or is it a necessary limit on potential collaborations? This type of insight would be particularly valuable in the manuscript's Housing Pathways and Themes section: Is there a causal or policy reason, for example, that all three agencies demonstrated significant interest in the Affordability pathway? Why, given the well-established links between homelessness, housing insecurity, and health, do those subpathways constitute a gap area in the overall portfolio?

I appreciated Figure 3—it’s a smart/effective way to present a lot of information in a visually consumable form.

Discussion: The discussion section identifies a mechanism that could address some of my questions above: defining common outcomes. Do the authors see common outcome definitions across agencies as a prerequisite framework for strengthening their shared research portfolio? Or do they see a role for the agencies' funded research itself to produce results that will inform the creation of those definitions?

Study Designs and Implications for Evidence Building: The authors find that studies focused in areas that lend themselves well to concrete interventions (homelessness, subsidized housing, housing remediation) are more likely to be intervention studies. By contrast, there are fewer intervention studies and more observational studies in areas where well-defined interventions that include fidelity models are difficult to design or deploy (built environment, neighborhood socioeconomic disadvantage, neighborhood risk characteristics, segregation). The analysis would be stronger if this distinction were called out explicitly; that would enable the authors to list a few potential approaches to addressing this gap that would be feasible both technically and within the context of HUD and HHS’s broader mandates.

To sum up: This paper is fairly explicitly an internal intervention in the funding priorities of three federal agencies. It has been written to meet the requirements of publication in a scientific journal interested in quantitative systemic reviews, and it is successful in that regard. But it would be more successful as an intervention published for an external audience if it included at least a bit more policy context to provide a lens into the institutions in which it seeks to intervene.

Reviewer #2: This is a terrific portfolio-wide analysis of NIH, CDC and HUD funding for homelessness and health research. It highlights some important gaps in current funding. The results are quite intelligible, and align with other recent reviews. I only note a few items for exploration that would enhance the contribution of this piece.

1) Methods: Why were 90% of the projects initially coded as being related to housing and health ultimately not coded by the research as being related to housing and health? It would help to have some idea of this. This rejection rate seems unusually high. I'm not asking for any methodological change, but I do think a table of reasons a paper was excluded could help. Very broad reasons. Or examples perhaps.

2) Methods: It’s not obvious why there are so many more NIH projects than HUD projects. I assume this relates largely to the fact that HUD and CDC don’t fund as much research, but it’s still surprising.

3) Results look great!

4) Discussion: The discussion section is perhaps unnecessarily long. It feels like the discussion might have more impact if it were synthetic in highlighting the key takeaways across all analyses rather than reviewing each analysis piece by piece.

5) I found it quite striking that homelessness, one of the most poorly funded areas of study, was also one of the few areas were intervention studies were dominant. I suspect that this results from hesitancy to conduct basic research on homelessness due to perceived difficulties of doing this work. As a homelessness researcher who has focused on basic research rather than interventions because there is little basis for knowing what works (other than housing), I believe this finding is incredibly important and worth highlighting in the discussion. I'd note recent reviews by Mosites et al. (https://academic.oup.com/aje/article/190/11/2432/6159692) and Richards (https://www.sciencedirect.com/science/article/pii/S2773065422000414) that highlight the gaps in basic research.

6) I think it should be noted as a limitation that given the overhwhelming dominance of NIH agencies, this analysis could have looked at specific funding Institutes, or at the scientific discipline. I suspect a number of findings such as the homelessness point noted above could be explained by disciplinary and institute biases towards particular issues or approaches.

Overall this paper will make a great contribution!

Randall

6)

6. PLOS authors have the option to publish the peer review history of their article (what does this mean?). If published, this will include your full peer review and any attached files.

Reviewer #1: No

Reviewer #2: **Yes: **Randall Kuhn

---

## [Author Response · Author response to Decision Letter 0]

8 Sep 2023

Dear Dr. Chang: 

I am submitting a revised version of “The intersection of health and housing: Analysis of the research portfolios of the National Institutes of Health, Centers for Disease Control and Prevention, and U.S. Department of Housing and Urban Development” (PONE-D-23-11897). My co-authors and I have carefully considered the reviewers’ comments. We appreciate the comments and made several changes to the manuscript in response. We believe that we have been responsive to the reviews and that the changes we made have improved the manuscript. I have uploaded two versions of the revised manuscript to the Editorial Manager system: one showing where changes were made, and one “clean” version. 

Below, we provide an itemized, point-by-point response to the reviewers’ comments. 

Thank you for the opportunity to submit this revision for consideration. Please let me know if you have any questions. My co-authors and I look forward to receiving your feedback on the revision and decision on the manuscript. 

Sincerely, 

Liberty Walton, MPH, CHES

NIH Office of Disease Prevention 

Email: Libery.Walton@nih.gov

Phone: 301-496-3979

PONE-D-23-11897: The intersection of health and housing: Analysis of the research portfolios of the National Institutes of Health, Centers for Disease Control and Prevention, and U.S. Department of Housing and Urban Development

RESPONSE TO THE REVIEWS

EDITOR:

Comment #1: Please ensure that your manuscript meets PLOS ONE's style requirements, including those for file naming. The PLOS ONE style templates can be found at https://journals.plos.org/plosone/s/file?id=wjVg/PLOSOne_formatting_sample_main_body.pdf and 

AUTHOR’S RESPONSE: Thank you for the links to these resources. I have reviewed our manuscript to ensure it meets PLOS ONE's style requirements.

Comment #2: In your Data Availability statement, you have not specified where the minimal data set underlying the results described in your manuscript can be found. PLOS defines a study's minimal data set as the underlying data used to reach the conclusions drawn in the manuscript and any additional data required to replicate the reported study findings in their entirety. All PLOS journals require that the minimal data set be made fully available. For more information about our data policy, please see http://journals.plos.org/plosone/s/data-availability.

AUTHOR’S RESPONSE: To meet the data sharing requirements of PLOS ONE, we have included our study’s minimal underlying data set as a Supporting Information file (S1 Table). 

Comment #3: We note that you have stated that you will provide repository information for your data at acceptance. Should your manuscript be accepted for publication, we will hold it until you provide the relevant accession numbers or DOIs necessary to access your data. If you wish to make changes to your Data Availability statement, please describe these changes in your cover letter and we will update your Data Availability statement to reflect the information you provide.

AUTHOR’S RESPONSE: To meet the data sharing requirements of PLOS ONE, we have included our study’s minimal underlying data set as a Supporting Information file (S1 Table).

Comment #4: Please review your reference list to ensure that it is complete and correct. If you have cited papers that have been retracted, please include the rationale for doing so in the manuscript text, or remove these references and replace them with relevant current references. Any changes to the reference list should be mentioned in the rebuttal letter that accompanies your revised manuscript. If you need to cite a retracted article, indicate the article’s retracted status in the References list and also include a citation and full reference for the retraction notice.

AUTHOR’S RESPONSE: We have reviewed our reference list to ensure that it is complete and correct. We did not cite papers that have been retracted. The only changes that have been made to our reference list was to add and remove references based on changes that we made to the manuscript text in response to the reviewer comments. 

Additional Editor Comments: The work presented is of high interest to the journal, and we encourage you to review the comments and suggestions to further enhance the manuscript.

 Please respond to the reviewers' comments, and in particular, please address the questions around the policy context of the research portfolios evaluated.

AUTHOR’S RESPONSE: We appreciate the positive feedback and comments from the editor. We have added language to the manuscript that we believe sufficiently addresses the questions around the policy context of the research portfolios evaluated.

Additional Editor Comments: While revising your submission, please upload your figure files to the Preflight Analysis and Conversion Engine (PACE) digital diagnostic tool, https://pacev2.apexcovantage.com/. PACE helps ensure that figures meet PLOS requirements. 

AUTHOR’S RESPONSE: Thank you for the link to this resource. We have uploaded our figure files to ensure our figures meet PLOS requirements. 

Reviewer 1: 

Overarching comments: The authors present a systematic review of research grants awarded to investigate the intersection of housing and health by three federal agencies over a period of five years. Their explicit research question is what types of research at the intersection of health and housing are currently being funded and what opportunities exist for additional research and collaboration (ie, what areas are underfunded, how can the agencies use grantmaking as a tool of collaboration). The implicit underlying question is whether public research dollars are being spent in the best possible way to improve the health of the U.S. population.

The approach employed in the review is to identify existing research by searching databases of studies funded by the NIH, the CDC, or HUD; classify each study based on themes, broad categories (“pathways”), populations of interest, types of study, etc; and quantify the studies based on those classifications to identify patterns and gaps in the funded portfolios.

The paper is somewhat unusual, in that the authors are employees of the funding organizations whose portfolios they evaluate. So, this is essentially an internal study that is seeking external publication, either in the name of transparency, in an attempt to influence the leadership or culture of the authors' home institutions, or simply to persuade the grant-seeking community to propose new projects that fill the gaps identified. Or all of the above.Based on this context, the introduction and discussion would be strengthened by a discussion of the role of public funding in general and of the three member agencies of the Health and Housing Group in particular in driving actionable research. What proportion of current health and housing policy research is funded by these agencies--and what effects do their grantmaking priorities have upon the research topics chosen across the policy research landscape writ large?

I would say this is the overall substance of my comments below: The authors observe patterns and gaps in a data set that are not random, but rather are the result of institutional policies and priorities enforced by human choices. The authors are, of course, in a slightly difficult position in that PLOS ONE accepts only systematic reviews that are based on reproducible quantitative analysis, so too much policy discussion would be inappropriate. Too little, however, fails to capture the causal mechanisms that influenced the patterns summarized in this paper, making it harder to fully grasp the most realistic levers available to close the portfolio gaps described. I recommend accepting the manuscript once some additional policy context is added--and once the data are made available. (The submitted manuscript specifies that the data "will only be available after acceptance." I'm not sure how this comports with the PLOS Data Policy.)

AUTHOR’S RESPONSE: Thank you for your thoughtful comments. The intent of our manuscript is to characterize the health and housing portfolios of the three federal agencies and to identify areas for further research and opportunities for cross-federal agency collaborative efforts to advance health and housing research. To address your comments, we added information on the mission and role in relation to health and housing of each agency to the Introduction. While there is not an official “health and housing” policy office at these agencies, we aim to organically grow collaborations and expand research in this area. We do not have data on all federal agencies that support health and housing, which we note in the limitations, but it is likely that these three agencies are the largest funders in their combined focus on health and housing research. Future analyses and collaborations may focus on additional agencies as we aim to expand our collaborations in the future. We note in the paper that we do not make policies, but we are able to build collaborations and inform external audiences on gap research areas. For external audiences, we aim to highlight research gap areas which may stimulate the submission of more research applications on health and housing. Most NIH projects are investigator-initiated, or unsolicited, so we believe this manuscript is an effective way to inform external audiences. For transparency and to meet the journal requirements, we added our data in a supplemental file.

Comment Regarding Conclusions: The authors determine that the existing portfolios could be strengthened if more studies were funded that a) sought to identify the specific causal mechanisms that link housing to health, b) evaluated the effectiveness of specific interventions, or c) otherwise provided concrete guidance as to how programs and policies could improve health by increasing the quality of, or access to, housing.

AUTHOR’S RESPONSE: We appreciate the accurate summary of the Conclusions. 

Comment Regarding Methods: Methods are well described. Categorization studies of this kind can never reach a level of 100% objectivity, so they can never be 100% reproducible, but the methods discussion convincingly indicates that subjectivity was minimized to the extent possible, and it provides sufficient information for other researchers to attempt to reproduce the results or to apply analogous methods to other agencies’ portfolios.

AUTHOR’S RESPONSE: We appreciate the positive feedback on the Methods. 

Comments Regarding Results: Following from my comment above about policy context, the analysis included in the manuscript's Results section would be strengthened if the Introduction included a brief overview of each agency’s policy role and agenda, which naturally informs its research interests and determines both the opportunities and the limits of the agencies’ ability to collaborate . The inspiration for this comment is Table 2, which demonstrates fairly clearly, for example, that HUD tends to fund research only into the specific types of housing resources that it creates or regulates, rather than the broader context in which those resources exist or intervene. (eg, 96% of HUD-funded research concerns the stability and quality of housing, while 0% concerns the health effects of homelessness and 2% concerns the built environment.) Does that narrow focus represent an opportunity, or is it a necessary limit on potential collaborations? This type of insight would be particularly valuable in the manuscript's Housing Pathways and Themes section: Is there a causal or policy reason, for example, that all three agencies demonstrated significant interest in the Affordability pathway? Why, given the well-established links between homelessness, housing insecurity, and health, do those subpathways constitute a gap area in the overall portfolio?

AUTHOR’S RESPONSE: Thank you for these thoughtful questions. As you noted, we fund research within our missions and mandates. This research can evaluate policy or be used externally to inform policy decisions. For example, through RFA-NR-23-003 (Evaluating the Impact of Pandemic Era related Food and Housing Policies and Programs on Health Outcomes in Health Disparity Populations), NIH supports research to evaluate policy, but does not create policies. One of the goals of our analysis was to understand what was being funded (or not funded); we did not explore the causal inferences as to specifically why projects were funded. In addition, as noted above, most NIH projects are investigator-initiated, allowing investigators to propose research that they believe will be most impactful. While we can only fund research within our specific missions and mandates, we can collaborate across agencies to leverage each other’s roles (e.g., data sharing) and coordinate and reduce duplication of efforts. To address your comments, we added information on the mission and role pertaining to health and housing of each agency to the Introduction. 

Comments Regarding Results: I appreciated Figure 3—it’s a smart/effective way to present a lot of information in a visually consumable form.

AUTHOR’S RESPONSE: Thank you for the positive feedback on Figure 3. 

Comments Regarding Discussion: The discussion section identifies a mechanism that could address some of my questions above: defining common outcomes. Do the authors see common outcome definitions across agencies as a prerequisite framework for strengthening their shared research portfolio? Or do they see a role for the agencies' funded research itself to produce results that will inform the creation of those definitions?

AUTHOR’S RESPONSE: This is a good suggestion to expand on common outcomes. We added the following to the Introduction as we describe the role of the Health and Housing Group, “A further goal of the health and housing working group has been to discuss the wide range of outcomes and exposures salient to health and housing research. Efforts to harmonize and prioritize health-related measurements are underway at NIH via the PhenX toolkit which already includes a few measures related to housing.”

Comments Regarding Discussion: Study Designs and Implications for Evidence Building: The authors find that studies focused in areas that lend themselves well to concrete interventions (homelessness, subsidized housing, housing remediation) are more likely to be intervention studies. By contrast, there are fewer intervention studies and more observational studies in areas where well-defined interventions that include fidelity models are difficult to design or deploy (built environment, neighborhood socioeconomic disadvantage, neighborhood risk characteristics, segregation). The analysis would be stronger if this distinction were called out explicitly; that would enable the authors to list a few potential approaches to addressing this gap that would be feasible both technically and within the context of HUD and HHS’s broader mandates.

AUTHOR’S RESPONSE: 

Thank you for suggesting this approach to strengthening our Discussion. We have added the following points to the Discussion, “As indicated by the funding patterns, some housing themes lend themselves to intervention studies, such as housing remediation, homelessness, and subsidized housing. By contrast, topics such as segregation and built environment can be more difficult to study with randomized intervention trials. A potential approach is ‘natural experiment’ study designs of a specific policy or environmental change, which could strengthen causal inference [26]. NIH has called for research using such approaches and funded several housing and health natural experiment studies [27, 28].”

Overarching comments: To sum up: This paper is fairly explicitly an internal intervention in the funding priorities of three federal agencies. It has been written to meet the requirements of publication in a scientific journal interested in quantitative systemic reviews, and it is successful in that regard. But it would be more successful as an intervention published for an external audience if it included at least a bit more policy context to provide a lens into the institutions in which it seeks to intervene.

AUTHOR’S RESPONSE: We appreciate your comments. In response to your comments, we added information on the mission and role in health and housing of each agency to the Introduction. We also more clearly defined the purpose of our analysis in the Introduction. The intent of our paper is to identify areas for further research and opportunities for cross-federal agency collaborative efforts to advance health and housing intervention research. 

Reviewer 2: 

Overarching comments: This is a terrific portfolio-wide analysis of NIH, CDC and HUD funding for homelessness and health research. It highlights some important gaps in current funding. The results are quite intelligible, and align with other recent reviews. I only note a few items for exploration that would enhance the contribution of this piece.

AUTHOR’S RESPONSE: We appreciate your positive feedback and comments. In response to your comments below, we have added language to the manuscript to enhance the contribution of this piece. 

Comment #1: Methods: Why were 90% of the projects initially coded as being related to housing and health ultimately not coded by the research as being related to housing and health? It would help to have some idea of this. This rejection rate seems unusually high. I'm not asking for any methodological change, but I do think a table of reasons a paper was excluded could help. Very broad reasons. Or examples perhaps.

AUTHOR’S RESPONSE: Thank you for pointing out that the reason for the high rejection rate was not clearly stated. We added examples to the Methods to explain why many projects were excluded. For example, many projects that were initially identified through our key word search for the term “housing” were excluded because they were basic research projects that mentioned “animal housing” in the abstract. 

Comment #2: Methods: It’s not obvious why there are so many more NIH projects than HUD projects. I assume this relates largely to the fact that HUD and CDC don’t fund as much research, but it’s still surprising.

AUTHOR’S RESPONSE: Yes, you are correct that NIH is the primary federal agency responsible for conducting and funding research. CDC and HUD fund both research and non-research programs, with much of the funding supporting demonstration programs. Funding for HUD research varies depending on the priorities of the administration. Additionally, while HUD funds many projects focused on housing, most of these projects do not have an explicit focus on health and therefore were not included in our analysis. We tried to address this comment in the Introduction by providing more details on the mission of each agency. 

Comment #3: Results look great!

AUTHOR’S RESPONSE: Thank you, we appreciate the positive feedback on the Results. 

Comment #4: Discussion: The discussion section is perhaps unnecessarily long. It feels like the discussion might have more impact if it were synthetic in highlighting the key takeaways across all analyses rather than reviewing each analysis piece by piece.

AUTHOR’S RESPONSE: Thank you for this suggestion. We re-organized the Discussion to align with the overarching themes that we identified at the beginning of the Discussion section and also streamlined the content. We believe that these changes have made a significant improvement to the focus and flow of the Discussion section. 

Comment #5: I found it quite striking that homelessness, one of the most poorly funded areas of study, was also one of the few areas were intervention studies were dominant. I suspect that this results from hesitancy to conduct basic research on homelessness due to perceived difficulties of doing this work. As a homelessness researcher who has focused on basic research rather than interventions because there is little basis for knowing what works (other than housing), I believe this finding is incredibly important and worth highlighting in the discussion. I'd note recent reviews by Mosites et al. (https://academic.oup.com/aje/article/190/11/2432/6159692) and Richards (https://www.sciencedirect.com/science/article/pii/S2773065422000414) that highlight the gaps in basic research.

AUTHOR’S RESPONSE: Thank you for this insightful feedback. We cited the articles that you mentioned and added the following to strengthen our Discussion, “Most studies in the portfolio across the three agencies were observational and non-experimental. However, the majority of studies on subsidized housing, homelessness, and housing remediations were intervention studies. Notably, only a third of studies on homelessness were observational. Recent research suggests that additional longitudinal, epidemiologic studies are needed to understand the causal mechanisms of poor health among persons experiencing homelessness [24, 25]”

Comment #6: I think it should be noted as a limitation that given the overhwhelming dominance of NIH agencies, this analysis could have looked at specific funding Institutes, or at the scientific discipline. I suspect a number of findings such as the homelessness point noted above could be explained by disciplinary and institute biases towards particular issues or approaches.

AUTHOR’S RESPONSE: While the focus of this analysis was to look broadly across the three agencies, this is a great idea for future analyses. We have noted this in the Limitations as follows, “In addition, since this portfolio was predominantly NIH-funded and NIH is made up of 27 Institutes, Centers, and Offices (ICOs), it may be more informative to conduct future analyses at the NIH ICO-level to better understand differences in funding across topics and study designs.”

Overarching comment: Overall this paper will make a great contribution!

AUTHOR’S RESPONSE: We thank Reviewer 2 for this positive feedback on our manuscript.

---

## [Decision Letter · Decision Letter 1]

18 Oct 2023

PONE-D-23-11897R1The intersection of health and housing: Analysis of the research portfolios of the National Institutes of Health, Centers for Disease Control and Prevention, and U.S. Department of Housing and Urban DevelopmentPLOS ONE

Dear Dr. Walton,

Thank you for submitting your manuscript to PLOS ONE. After careful consideration, we feel that it has merit but does not fully meet PLOS ONE’s publication criteria as it currently stands. Therefore, we invite you to submit a revised version of the manuscript that addresses the points raised during the review process.

One of the reviewers has provided a few additional comments to improve your paper further. I encourage you to carefully consider the reviewer's feedback and address each point raised in your revision.

We look forward to receiving your revised manuscript.

Kind regards,

Dong Liu, Ph.D.

Academic Editor

PLOS ONE

Journal Requirements:

Reviewers' comments:

Reviewer's Responses to Questions

**Comments to the Author**

1. If the authors have adequately addressed your comments raised in a previous round of review and you feel that this manuscript is now acceptable for publication, you may indicate that here to bypass the “Comments to the Author” section, enter your conflict of interest statement in the “Confidential to Editor” section, and submit your "Accept" recommendation.

Reviewer #3: All comments have been addressed

Reviewer #4: All comments have been addressed

2. Is the manuscript technically sound, and do the data support the conclusions?

Reviewer #3: Yes

Reviewer #4: Yes

3. Has the statistical analysis been performed appropriately and rigorously? 

Reviewer #3: Yes

Reviewer #4: Yes

4. Have the authors made all data underlying the findings in their manuscript fully available?

Reviewer #3: Yes

Reviewer #4: No

5. Is the manuscript presented in an intelligible fashion and written in standard English?

Reviewer #3: Yes

Reviewer #4: Yes

6. Review Comments to the Author

Reviewer #3: This is a very interesting paper. The focus on an analysis of NIH, CDC and HUD funding for housing and health research. After addressing two reviewers comments, the current introduction and literature review are comprehensive and up to date and the research purpose is clearly stated. The methods and results are explained and discussed well. I do not have any comments for the authors.

Reviewer #4: This article delves into the realm of home and health research, leveraging data from prominent research projects by NIH, HUD, and CDC. I find that the article presents a well-defined framework and holds significant implications for housing intervention policies and the enhancement of residents' well-being. In general, the author has effectively addressed the comments derived from the first round of review. However, in order to enhance the prospects of publication, I have additional several comments:

1. The literature review concerning home and health should be expanded. However, there is insufficient summary of relevant work.

2. It would be beneficial to incorporate a discussion of globally pertinent policies related to home and health in the section of Introduction.

3. In the methodology section, it would be advantageous to provide an in-depth explanation of how the NIH, HUD, and CDC research projects can be harnessed to gain deeper insights into the dynamics of home and health.

4. The current reliance on basic statistical analysis may be perceived as somewhat simplistic. If feasible, the authors might consider constructing mathematical models to elucidate the underlying mechanisms, or provide a more comprehensive discussion regarding the suitability of these basic statistical analyses within the context of this article.

7. PLOS authors have the option to publish the peer review history of their article (what does this mean?). If published, this will include your full peer review and any attached files.

Reviewer #3: No

Reviewer #4: No

---

## [Author Response · Author response to Decision Letter 1]

1 Dec 2023

PONE-D-23-11897R1: The intersection of health and housing: Analysis of the research portfolios of the National Institutes of Health, Centers for Disease Control and Prevention, and U.S. Department of Housing and Urban Development

RESPONSE TO THE REVIEWS

Reviewer 3: 

Overarching comments: This is a very interesting paper. The focus on an analysis of NIH, CDC and HUD funding for housing and health research. After addressing two reviewers comments, the current introduction and literature review are comprehensive and up to date and the research purpose is clearly stated. The methods and results are explained and discussed well. I do not have any comments for the authors.

AUTHOR’S RESPONSE: Thank you very much for your positive feedback. We appreciate your review. 

Reviewer 4: 

Overarching comments: This article delves into the realm of home and health research, leveraging data from prominent research projects by NIH, HUD, and CDC. I find that the article presents a well-defined framework and holds significant implications for housing intervention policies and the enhancement of residents' well-being. In general, the author has effectively addressed the comments derived from the first round of review. However, in order to enhance the prospects of publication, I have additional several comments:

AUTHOR’S RESPONSE: Thank you very much for your supportive review. In response to your comments, we have included some revisions, which we believe address the comments and further strengthen the manuscript. 

Comment #1: The literature review concerning home and health should be expanded. However, there is insufficient summary of relevant work.

AUTHOR’S RESPONSE: Thank you very much for the suggestion that the literature review concerning home and health should be expanded. We have expanded the Introduction and referenced additional literature on housing and health. We note that Reviewer 3 commented that, “After addressing two reviewer’s comments, the current introduction and literature review are comprehensive and up to date…,” and much of this literature is also referenced in the Discussion. Given this paper is a descriptive review of funding, rather than a comprehensive literature review, we hope these new additions are satisfactory.

Comment #2: It would be beneficial to incorporate a discussion of globally pertinent policies related to home and health in the section of Introduction.

AUTHOR’S RESPONSE: Thank you for this suggestion. We highlight housing and health guidelines for the United States and the World Health Organization in the Introduction, and we added mention of recent housing initiatives in Canada and the UK. We also added additional information in the Discussion to reinforce the global relevance. 

Comment #3: In the methodology section, it would be advantageous to provide an in-depth explanation of how the NIH, HUD, and CDC research projects can be harnessed to gain deeper insights into the dynamics of home and health.

AUTHOR’S RESPONSE: Thank you for this suggestion. We added text to the Introduction that indicates that the results of NIH, HUD, and CDC research both inform US policy and subsequent research efforts and contribute to public health recommendations. We also note in the Discussion section that the results offer critical evidence for program and policy decisions. The Methods section links to the supplemental file containing all the project abstracts which readers can read for more in-depth insights on the research supported by these three agencies. 

Comment #4: The current reliance on basic statistical analysis may be perceived as somewhat simplistic. If feasible, the authors might consider constructing mathematical models to elucidate the underlying mechanisms, or provide a more comprehensive discussion regarding the suitability of these basic statistical analyses within the context of this article.

AUTHOR’S RESPONSE: Thank you for this feedback. The main objective of this manuscript is to describe the health and housing research portfolio across NIH, HUD, and CDC to identify research gaps and suggest future avenues of research. We’ve added language to the Methods to communicate that little is known about the combined agencies health and housing portfolios, and thus a descriptive analysis is a necessary step towards identifying areas for future research. While it may be perceived as somewhat simplistic, it is an approach that can commonly be found in the literature, including a 2021 PLOS ONE article (Shams-White, et al. 2021). We agree that mathematical models could be informative for future analyses, and we discuss this in the Limitations section.

---

## [Decision Letter · Decision Letter 2]

27 Dec 2023

The intersection of health and housing: Analysis of the research portfolios of the National Institutes of Health, Centers for Disease Control and Prevention, and U.S. Department of Housing and Urban Development

PONE-D-23-11897R2

Dear Dr. Walton,

We’re pleased to inform you that your manuscript has been judged scientifically suitable for publication and will be formally accepted for publication once it meets all outstanding technical requirements.

Kind regards,

Dong Liu, Ph.D.

Academic Editor

PLOS ONE

Reviewers' comments:

Reviewer's Responses to Questions

**Comments to the Author**

1. If the authors have adequately addressed your comments raised in a previous round of review and you feel that this manuscript is now acceptable for publication, you may indicate that here to bypass the “Comments to the Author” section, enter your conflict of interest statement in the “Confidential to Editor” section, and submit your "Accept" recommendation.

Reviewer #4: All comments have been addressed

2. Is the manuscript technically sound, and do the data support the conclusions?

Reviewer #4: Yes

3. Has the statistical analysis been performed appropriately and rigorously? 

Reviewer #4: Yes

4. Have the authors made all data underlying the findings in their manuscript fully available?

Reviewer #4: Yes

5. Is the manuscript presented in an intelligible fashion and written in standard English?

Reviewer #4: Yes

6. Review Comments to the Author

Reviewer #4: The authors addressed the comments and suggestions I provided, and the quality of the article has greatly improved. I believe it can be published on PLOS ONE.

7. PLOS authors have the option to publish the peer review history of their article (what does this mean?). If published, this will include your full peer review and any attached files.

Reviewer #4: No

---

## [Editor Report · Acceptance letter]

19 Jan 2024

PONE-D-23-11897R2 

PLOS ONE

Dear Dr. Walton, 

I'm pleased to inform you that your manuscript has been deemed suitable for publication in PLOS ONE. Congratulations! Your manuscript is now being handed over to our production team.

Kind regards, 

on behalf of

Professor Dong Liu 

Academic Editor

PLOS ONE